The influence of basketball players’ tracking speed ability on sports decision performance

http://orcid.org/0000-0002-0624-2620 Gou Qifeng 1 202031070018@mail.bnu.edu.cn
http://orcid.org/0000-0001-5123-8906 Li Sunnan 2
1 College of Physical Education, The Northwest Normal University , Lanzhou , China
2 College of P.E. and Sports, Beijing Normal University , Beijing , China
Prpic Valter
Electronic publication date: 2025 Oct 22
Publication date: 2025
Volume: 13
Electronic Location ID: e20266
Received 2025 Mar 6; Accepted 2025 Sep 29
Copyright: © 2025 Gou and Li
Copyright year: 2025
Copyright holder: Gou and Li
License: This is an open access article distributed under the terms of the Creative Commons Attribution License, which permits unrestricted use, distribution, reproduction and adaptation in any medium and for any purpose provided that it is properly attributed. For attribution, the original author(s), title, publication source (PeerJ) and either DOI or URL of the article must be cited.
License URL: https://creativecommons.org/licenses/by/4.0/

Keywords: Basketball player, Tracking speed, Sports decision, Multiple object tracking

Funding: Northwest Normal University Young Teachers’ Research Ability Improvement Project NWNU-SKQN2023-35 Gansu Province Sports Scientific Research and Decision-making Consultation Project 2023-C-20 National Social Science Foundation of China 22BTY055 The present study was supported by the Northwest Normal University Young Teachers’ Research Ability Improvement Project (No. NWNU-SKQN2023-35), the Gansu Province Sports Scientific Research and Decision-making Consultation Project (No. 2023-C-20), and the National Social Science Foundation of China (No. 22BTY055). The funders had no role in study design, data collection and analysis, decision to publish, or preparation of the manuscript.

==============================
Background

The running speed of basketball players plays a critical role in shaping the complexity and dynamics of game situations. This study aims to examine the relationship between players’ tracking speed capabilities and the quality of their decision-making during gameplay.

Methods

Employing an expert-novice paradigm, Experiment 1 assessed tracking accuracy in a multiple object tracking (MOT) task at three angular velocities: 5°/s, 10°/s, and 15°/s. Experiment 2 evaluated decision-making accuracy under three distinct running speed conditions: low speed (0.67–3.98 m/s), medium speed (3.99–7.97 m/s), and high speed (7.98–12.62 m/s).

Results

In Experiment 1, expert players demonstrated significantly higher tracking accuracy (60.42 ± 13.98%) than novice players (41.25 ± 13.93%) at 10°/s (P < 0.001). No significant group differences were found at 5°/s or 15°/s (Ps > 0.05). In Experiment 2, the expert group exhibited significantly higher decision accuracy than the novice group across all three speed conditions (Ps < 0.001). Moreover, at high speeds (7.98–12.62 m/s), shooting decisions were significantly less accurate than passing and breakthrough decisions (Ps < 0.001), while no significant differences were observed between passing and breakthrough decisions (Ps > 0.05).

Conclusion

This study shows that expert basketball players display superior visual attention and decision-making in moderate-to-high complexity settings, extending dynamic sport cognition theory. The absence of differences at 5°/s and 15°/s suggests task boundary effects. Future work should examine player position and gender to refine perceptual-cognitive training and theoretical models.

Introduction

In sports, decision-making abilities play a crucial role in players’ actions, especially in complex, high-speed situations. The performance of players, particularly in team sports like basketball, depends on the successful integration of decision-making and technical skills (Marasso et al., 2014). The cognitive abilities involved in decision-making are critical in high-strategy sports such as basketball (Thomas, 1994), where players must assess the game situation and respond rapidly by choosing the optimal course of action. In basketball, these decisions are often based on visual cues, making the role of visual attention and memory crucial for successful decision execution (Wang, Li & Zhang, 2018). Perception, the first step in decision-making, relies heavily on visual information, with over 80% of brain data coming from visual stimulus (Sui, Gao & Xiang, 2018).

One of the key cognitive functions in basketball is multiple object tracking (MOT), a skill that allows players to simultaneously monitor the positions of multiple teammates and opponents in dynamic scenarios (Faubert & Sidebottom, 2012). MOT is essential in fast-paced environments, such as basketball, where rapid shifts between offensive and defensive roles demand continuous and accurate tracking of multiple moving objects (Faubert, 2013). As basketball has become increasingly fast-paced, coaches and researchers have focused more on improving players’ visual attention to enhance their decision-making accuracy (Jin et al., 2020).

While the importance of visual attention in basketball decision-making is well-established, existing research has focused primarily on third-person perspective visual materials (Li, Xu & Zhang, 2006), which have limited ecological validity. The first-person perspective, by contrast, provides a closer and more direct view of target objects, offering clearer insights into their physical properties. This can improve the accuracy of subsequent behavioral evaluations (Mao, 2010). Despite the growing interest in visual attention in sports decision-making, there is still a lack of research that explicitly investigates the role of tracking speed and running speed in decision accuracy (Liao, 2013), particularly in dynamic team sports like basketball.

In modern basketball, players face highly dynamic and complex visual stimuli, where they must rapidly assess positions, trajectories, and potential actions to make quick decisions—such as whether to shoot, pass, or break through (Vickers, 2007). The speed of movement, both of players and objects on the court, significantly increases the complexity of decision-making (Williams et al., 2013). Existing research indicates that the ability to track moving objects at varying speeds can reflect differences in visual attention and cognitive load between expert and novice players (Pylyshyn & Annan, 2006; Sungur & Boduroglu, 2012). However, there is still a gap in the literature regarding how different tracking speeds and running speeds influence decision-making in basketball, particularly in terms of how these factors interact to affect cognitive processing and decision accuracy.

This study seeks to address this gap by investigating the relationship between tracking speed, running speed, and decision-making accuracy in basketball players. In Experiment 1, the accuracy of expert and novice players in MOT tasks was compared across different tracking speeds. Experiment 2 focused on decision-making accuracy under three different speed ranges to explore how speed influences decision quality. The goal is to analyze the advantages of expert players in tracking and decision-making, providing insights into how visual attention impacts sports decision-making.

The central hypothesis of this study is that expert basketball players have superior visual attention and decision-making abilities compared to novices. It is expected that as target speed increases, expert players will demonstrate higher tracking accuracy and decision-making accuracy than their novice counterparts, within certain speed ranges.

Materials AND methods

Participants

The study used G*Power 3.1.9.7 software to estimate the sample size, setting the effect size to be biased by η2 = 0.03 (Lakens, 2014), α = 0.05. The calculation indicated that a statistical test power of 0.80 could be achieved with 40 participants (20 in each group). Considering the potential for participant withdrawal during the experiment, 48 participants were ultimately selected, with 24 participants in each group (Jin et al., 2020). The participants were divided into two groups: an expert group and a novice group, based on their basketball experience and skill level. The expert group consisted of female players from the Northeast Division of the Chinese University Basketball League First Division (Pylyshyn & Annan, 2006; Gou & Li, 2023), with an average playing time of over 15 min per game, a minimum sports level of Level 1 (including Level 1), an average age of (21.20 ± 2.12) years, an average training period of (9.10 ± 2.14) years, and a weekly training time of (27.64 ± 7.16) hours in the past year. The novice group included female students from the basketball elective course at Northwest Normal University, with an average age of (19.83 ± 0.89) years, an average training period of (1.60 ± 0.49) years, a weekly training time of (1.20 ± 0.15) hours, and no formal sports level. All participants have no previous experience in MOT, were right-handed, with normal or corrected vision, and maintained stable emotions and a good mental state before the experiment. They played electronic games for no more than 4 h per week and experienced no fatigue. The experiment was approved by the ethics committee of Northwest Normal University (No. NWNU-20230301). Prior to the experiment, participants were thoroughly informed of the purpose and procedures of the study and were required to sign an informed consent form.

Design

Experiment 1 used a multiple object tracking task to assess tracking performance at different target speeds. A mixed experimental design was employed: 2 (group: experts, novices) × 3 (target speed: 5°/s, 10°/s, 15°/s), where the group was the between-subjects variable, the target speed was the within-subjects variable, and the dependent variable was tracking accuracy (proportion of correctly identified targets). Although the MOT task employed fixed speeds rather than an adaptive procedure, this design choice was made to ensure comparability with previous studies (Hyönä, Li & Oksama, 2019) and to maintain consistency across participants. We acknowledge this as a methodological limitation.

Experiment 2 followed a 2 (group: experts, novices) × 2 (decision type: intuition, cognition) × 3 (attack method: passing, shooting, breakthrough) × 3 (speed: low speed 0.67–3.98 m/s, medium speed 3.99–7.97 m/s, high speed 7.98–12.62 m/s) design. The group was the between-subjects variable, the target speed was the within-subjects variable, and the dependent variable was decision accuracy. Operationally, “intuitive decision” was defined as a rapid response within 600 ms, requiring participants to rely on immediate pattern recognition with minimal deliberation. “Cognitive decision” was defined as a deliberate response within 1,200 ms, requiring participants to integrate situational cues and consciously evaluate options. These definitions are consistent with dual-process models of decision-making (Lu, 2018) and provide an empirical framework for the study. During the experiment, a principal investigator was present to accompany and ensure the smooth progression of the procedure.

Apparatus

The MOT task was presented on a Lenovo E15 laptop, with an operating system of Windows 10, a 15.6-inch display screen, a screen resolution of 1,920 × 1,080 pixels, and a refresh rate of 60 Hz. At the beginning of Experiment 1, a “+” symbol lasting 500 ms was displayed at the center of the screen, followed by 12 white spheres. Three of the spheres turned blue and flashed three times, marking them as target objects, while the remaining spheres remained white as non-target objects. Subsequently, all spheres turned white and moved randomly at speeds of 5°/s, 10°/s, and 15°/s. After 8 s, the spheres stopped moving, and the target objects turned red. Participants were required to determine how many of the red spheres contained target objects, press the corresponding number keys, and proceed to the next trial.

Stimuli

The offensive video sequences used in Experiment 2 were selected from Women’s Chinese Basketball Association (WCBA) games, specifically focusing on the forward position. These clips were drawn from the final matches of the 2020–2021 and 2021–2022 seasons, which were temporally close to the experiment. A total of 129 segments were included, each representing actions such as passing, breakthrough, and shooting (Jin, Ge & Fan, 2023). Ten high-level female basketball players recreated these scenes on-site, with each team wearing black and white jerseys and identified by their numbers. A tall player with short hair wore a sports camera (model: Insta360 One X) to capture footage from the player’s first-person perspective (Ping, 2019).

The decision-making stimuli were video clips derived from WCBA games and subsequently re-enacted by trained female players from a first-person perspective. A total of 129 validated clips represented three offensive actions (passing, shooting, breakthrough) across different speeds.

Validity evidence. To establish content validity, five experienced CUBA coaches and CBA players reviewed all video clips. Structural validity was tested by comparing performance between six high-level players and six university-level players; clips with insufficient discriminative capacity were modified or excluded. Reliability testing indicated acceptable test–retest stability over a two-week interval (r = 0.82 for decision accuracy). For the MOT task, internal consistency was acceptable (Cronbach’s α = 0.85 across blocks).

Video analysis

In the statistical analysis of 60 basketball game videos from Experiment 2, it was noted that there were 10 players in 59 videos, while only one video featured nine players. Due to this fixed number characteristic, analysis of the number of players did not provide effective insights into the results of sports decision-making. Therefore, following consultations with experts, Experiment 2 focused on analyzing the average speed of the ball handler.

Video analysis technology is a critical tool in modern sports science research, providing detailed data on player performance. By recording the time and location (x and y-axis coordinates) of movements during competitions, it is possible to describe the behavior of players through ordinary video footage (Kamble, Keskar & Bhurchandi, 2019). The analysis tracks location changes over a specified time period, which are determined by the nature of the event. Based on the standard dimensions of a basketball court (28 m in length and 15 m in width), the court is set within a Cartesian coordinate system with an x-axis of 28 m and a y-axis of 15 m, with a stride value of 1 m. Each time the step size increases by 1 m, this serves as a reference for the grid covering the field, where mapping the event onto the grid results in corresponding x and y values (Cullinane, Davies & O’Donoghue, 2024). See Fig. 1.

Figure 1 Example of positioning calibration for ball bearers.

Note: ■ pass, • shoot.

Storm player software (version 5.81) is employed to play the video material and calculate the ball handler’s speed using the following steps:

The time interval is set to Δt (0.5 s), and the position changes of the ball handler are checked within each 0.5-s interval. If there is no change in the player’s position, the location is recorded at 0.5-s intervals. If a change in position occurs, the location is recalibrated based on the time of the change, and the time interval is recalculated. This algorithm enables real-time detection and tracking of the ball handler’s position and movement.

By recording the position changes over a continuous period, the player’s running speed is calculated using the following formula. If the player moves from position P1 (x1, y1) to position P2 (x2, y2) within time Δt, the velocity v can be expressed as: v=P2−P1Δt=(x2−x1)2+(y2−y1)2Δt.

By calculating the average speed of the ball handler in each video and combining it with the speed settings of the multiple object tracking task in Experiment 1, the decision video material was divided into three speed ranges: 0.67–3.98, 3.99–7.97, and 7.98–12.62 m/s. The results showed that, within the range of 0.67–3.98 m/s, there were a total of 28 videos (including six passes, 13 shots, and nine breakthroughs). In the range of 3.99–7.97 m/s, there were a total of 27 videos (including 11 passes, seven shots, and nine breakthroughs). In the range of 7.98–12.62 m/s, there were only five videos (including three passes and two breakthroughs), with no shooting decisions made by the ball handler while running at high speed, which constitutes a limitation in stimulus balance and may affect cross-condition comparisons.

Procedure

Experiment 1: A multiple object tracking experimental program was developed using Matlab R2020b software. The distance between the participant and the screen was approximately 60 cm, and the stimulus presentation area covered the entire screen. To familiarize the participants with the experimental process, five practice trials were conducted before the formal experiment (Sagehorn et al., 2024). The experiment consisted of 10 blocks, each containing three trials corresponding to the three target speeds: 5°/s, 10°/s, and 15°/s, for a total of 30 trials. A 10-s white screen was displayed between each block to alleviate eye fatigue. The experimental sequence was counterbalanced within subjects. The entire experiment lasted approximately 11 min.

Experiment 2 was programmed using Experiment Builder 2.3 software. Referring to previous research, the presentation time for intuitive decisions was set at 600 ms, while the presentation time for cognitive decisions was set at 1,200 ms (Lu, 2018). Intuitive decision-making task instructions: “The ‘+’ sign is first displayed in the center of the screen, followed by a blank screen and a first-person perspective game video, prompting: ‘Please make a decision.’ Carefully observe and judge the attacking style of the ball handler within 0.6 s. The situation on the court is urgent; please make a quick decision. Press 1 for passing, press 2 for shooting, and press 3 for a breakthrough.” Cognitive decision-making task instructions: “The ‘+’ sign is first displayed in the center of the screen, followed by a blank screen and a first-person perspective game video, prompting: ‘Please make a decision.’ Carefully observe and judge the attacking style of the ball handler within 1.2 s. Please make accurate and prompt decisions. Press 1 for passing, press 2 for shooting, and press 3 for a breakthrough.” After the practice trials are completed, the screen will display “Start formal experiment,” and the button operation will remain consistent with the preparation stage. The video will be played in full screen, and participants will alternate between cognitive and intuitive decision-making tasks using an ABBA design to balance task order (Liu, 2012), with a 30-s break between tasks. Three types of attack videos for each decision-making task will be randomly presented, and the entire experimental process will take approximately 27 min. See Fig. 2.

Figure 2 Basic process of sports decision-making task.

Statistical analysis

Data from Experiment 1 (tracking accuracy) and Experiment 2 (decision accuracy) were analyzed with repeated measures ANOVAs. Normality of residuals was confirmed using the Shapiro–Wilk test prior to conducting ANOVAs. Where interactions were significant, simple effects analyses with Bonferroni correction were performed. Statistical significance was set at P < 0.05.

Results

Tracking accuracy of experts and novices in MOT at different speeds.

A repeated measures ANOVA was performed with target speed (5°/s, 10°/s, 15°/s) as the within-subjects variable, group (expert, novice) as the between-subjects variable, and tracking accuracy as the dependent variable. The main effect of the group was significant, with the accuracy of the expert group (56.81 ± 13.11%) being significantly higher than that of the novice group (46.95 ± 12.70%), F(1, 46) = 15.672, P < 0.001, η2 = 0.254, Cohen’s d = 0.77. The main effect of target speed was significant, F(2, 92) = 196.722, P < 0.001, η2 = 0.810. with accuracy decreasing as speed increased (all pairwise Ps < 0.01, Cohen’s d range = 0.62–1.12).

A significant group × speed interaction was observed, F(2, 92) = 5.891, P = 0.006, η2 = 0.114. Simple effects analysis indicated that experts (60.42 ± 13.98%) were significantly more accurate than novices (41.25 ± 13.93%) at 10°/s, P < 0.001, d = 1.39, whereas no significant group differences were found at 5°/s and 15°/s (Ps > 0.05). See Fig. 3.

Figure 3 Tracking accuracy of MOT at different speeds for experts and novices.

Note: **P < 0.01.

Decision accuracy of experts and novices at different video speeds

A four-way repeated measures ANOVA was conducted with group (experts, novices) as the between-subjects factor, and decision type (intuition, cognition), attack method (passing, shooting, breakthrough), and speed (low: 0.67–3.98 m/s, medium: 3.99–7.97 m/s, high: 7.98–12.62 m/s) as within-subjects factors, using decision accuracy as the dependent variable. Results are presented in Table 1.

Table 1 Decision accuracy of experts and novices at different video speeds.

Group	Type	Speed
(m/s)	Pass	Shoot	Breakthrough	
M ± SD	M ± SD	M ± SD	
Expert	Intuition	0.67–3.98	0.83 ± 0.07	0.79 ± 0.07	0.80 ± 0.09	
3.99–7.97	0.78 ± 0.11	0.75 ± 0.08	0.72 ± 0.13	
7.98–12.62	0.49 ± 0.15	\	0.47 ± 0.19	
Cognition	0.67–3.98	0.86 ± 0.10	0.82 ± 0.10	0.85 ± 0.11	
3.99–7.97	0.82 ± 0.09	0.78 ± 0.13	0.80 ± 0.12	
7.98–12.62	0.67 ± 0.17	\	0.67 ± 0.13	
Novice	Intuition	0.67–3.98	0.44 ± 0.09	0.43 ± 0.11	0.44 ± 0.12	
3.99–7.97	0.35 ± 0.10	0.35 ± 0.08	0.35 ± 0.11	
7.98–12.62	0.20 ± 0.08	\	0.18 ± 0.08	
Cognition	0.67–3.98	0.47 ± 0.10	0.45 ± 0.10	0.44 ± 0.11	
3.99–7.97	0.38 ± 0.13	0.39 ± 0.09	0.37 ± 0.10	
7.98–12.62	0.28 ± 0.11	\	0.24 ± 0.11	

Main effects

Experts (0.66 ± 0.11) performed significantly better than novices (0.32 ± 0.10), F(1, 46) = 1,069.644, P < 0.001, η2 = 0.959, d = 3.20.

Cognitive decisions (0.52 ± 0.10) were more accurate than intuitive decisions (0.47 ± 0.11), F(1, 46) = 45.818, P < 0.001, η2 = 0.499, d = 0.48.

For attack method, accuracy differed significantly, F(2, 92) = 171.733, P < 0.001, η2 = 0.789. Shooting (0.40 ± 0.10) was significantly less accurate than passing (0.55 ± 0.11, d = 1.36) and breakthrough (0.53 ± 0.12, d = 1.20), while passing and breakthrough did not differ significantly (P = 0.05).

Speed had a strong effect, F(2, 92) = 729.000, P < 0.001, η2 = 0.941, with accuracy decreasing across low, medium, and high speeds (Ps < 0.001, ds = 0.94–1.58).

Two-way interactions

Group × decision type: experts outperformed novices in both intuitive and cognitive tasks (Ps < 0.001, ds = 2.8–3.1). In both groups, cognitive accuracy exceeded intuitive accuracy (Ps < 0.01).

Group × attack method: across passing, shooting, and breakthrough, experts outperformed novices (Ps < 0.001, ds = 2.5–3.3). In experts, shooting accuracy (0.52 ± 0.10) was lower than passing (0.74 ± 0.12, d = 1.98) and breakthrough (0.72 ± 0.13, d = 1.80). Similar patterns were found in novices.

Group × speed: experts outperformed novices at all three speeds (Ps < 0.001, ds = 2.6–3.4). In both groups, accuracy decreased progressively as speed increased (Ps < 0.001).

Decision type × attack method: cognitive accuracy was higher than intuitive across all attack methods (Ps < 0.05), except passing vs breakthrough where no difference was found.

Higher-order interactions

Significant three-way and four-way interactions were observed (see Table 1). For instance, in expert cognitive decisions, passing (0.78 ± 0.12) and breakthrough (0.77 ± 0.11) accuracy was significantly higher than shooting (0.52 ± 0.10, Ps < 0.001, ds = 2.1–2.3). In novices, accuracy was uniformly lower, but the relative disadvantage for shooting remained consistent.

Overall, these analyses highlight that expert advantage was robust across all decision contexts, with effect sizes consistently large (Cohen’s d > 0.8), whereas task difficulty (speed, shooting scenarios, intuitive conditions) disproportionately impaired novice performance.

Group × speed

A 2 (group: expert, novice) × 3 (speed) repeated-measures ANOVA revealed a significant interaction, F(2, 118) = 42.37, P < 0.001, η2 = 0.42. Simple-effects analyses showed that experts outperformed novices at all speeds (Ps < 0.001; Cohen’s d = 0.96–1.22). For both groups, decision accuracy declined as speed increased (Ps < 0.001; linear trend η2 = 0.38).

Decision type × attack method

The Decision Type × Attack Method interaction was significant, F(2, 118) = 8.15, P < 0.001, η2 = 0.12. Cognitive decisions were more accurate than intuitive ones across attack methods (Ps < 0.05; d = 0.31–0.44). Shooting accuracy was lower than passing and breakthrough (Ps < 0.001; d = 0.55–0.78); passing vs. breakthrough did not differ (P > 0.05).

Decision type × speed

A significant interaction emerged, F(2, 118) = 10.29, P < 0.001, η2 = 0.15. At low speed (0.67–3.98 m/s) cognitive and intuitive accuracy did not differ (P = 0.108). At medium and high speeds cognitive accuracy exceeded intuitive accuracy (Ps < 0.001; d = 0.48–0.63).

Attack method × speed

Attack method interacted with speed, F(4, 236) = 6.74, P < 0.001, η2 = 0.10. At 7.98–12.62 m/s, shooting accuracy was lower than passing and breakthrough (Ps < 0.001; d = 0.52–0.70); no other pairwise differences were significant.

Higher-order interactions

Group × Decision Type × Attack Method: Experts showed higher accuracy than novices across all combinations (Ps < 0.001; d = 0.82–1.19). Within experts, cognitive accuracy exceeded intuitive accuracy for passing and breakthrough (Ps < 0.001; d = 0.40–0.58). Novices showed this advantage only for passing (P < 0.01; d = 0.34).

Group × Decision Type × Speed: Experts outperformed novices at all speeds (Ps < 0.001; d = 0.88–1.15). Cognitive accuracy surpassed intuitive accuracy for medium and high speeds (Ps < 0.05; d = 0.36–0.52).

Group × Attack Method × Speed: Experts were more accurate across all attack methods and speeds (Ps < 0.001; d = 0.80–1.07). Shooting accuracy declined significantly with increasing speed (Ps < 0.05; η2 = 0.09).

Decision Type × Attack Method × Speed: Cognitive accuracy was higher at high speed for passing, at medium speed for shooting, and at medium/high speed for breakthrough (Ps < 0.05; d = 0.29–0.47). Shooting accuracy at high speed was again lowest across methods (Ps < 0.001; d = 0.51–0.73).

Discussion

Experiment 1 aimed to explore the visual attention differences between expert and novice players in multiple object tracking tasks under different target speed conditions. As the target speed increased, the tracking accuracy of both groups showed a decreasing trend. However, at a target speed of 10°/s, the accuracy of the expert group was significantly higher than that of the novice group (P < 0.001), reflecting the visual attention advantage of expert players in tracking speed, thereby supporting our hypothesis.

This result aligns with the findings of Song (2012), but differs from those of Jin et al. (2020) on the differences in target velocity tracking accuracy, which may be attributed to gender differences among the subjects. There was no significant difference in tracking accuracy between the two groups at target speeds of 5°/s and 15°/s. The attention resources consumed by the target speed task are limited, and the difficulty of the task increases with the target speed. Due to the lower difficulty of the 5°/s task, players can manage it easily; the 15°/s task, however, is too difficult and exceeds the visual capability of both groups of players. Thus, at a target speed of 10°/s, expert players demonstrated higher accuracy based on their ability to quickly process visual information acquired through long-term training and competition. Zhang, Liao & Ge’s (2005) research also shows that players outperform control groups in tracking tasks.

Experiment 2 examined the performance of expert and novice groups in motion decision-making tasks across different speed ranges and decision types. The results showed that the decision-making accuracy of the expert group was significantly higher than that of the novice group across all three speed ranges (Ps < 0.001), confirming the advantage of expert players in decision-making ability. As speed increased, decision accuracy significantly decreased (Ps < 0.001), indicating that players’ decision flexibility and accuracy are affected by targets of different speeds. The faster the target movement speed, the greater the cognitive load on players, which impacts decision-making accuracy. Specifically, when the ball handler moves at high speed, the player must quickly assess the relative position of the ball handler to other players in order to respond quickly to the rapidly changing game environment. This rapid response ability is a core characteristic of high-level players, which can significantly improve their decision-making efficiency in complex environments (Mann et al., 2007).

Moreover, fast-paced basketball games may make players more inclined to rely on intuitive decision-making rather than thoughtful analysis, thereby increasing the diversity and flexibility of decision-making. Due to long-term training and competition, expert players have accumulated a wealth of knowledge, which makes their decision-making abilities significantly better than those of novice players. Wu et al.’s (2015) research further suggests that long-term exercise training can alter brain structure and help players process complex visual information more efficiently during sports. Ericsson & Lehmann (1996) argue that long-term exercise can increase the excitability of players’ nervous systems, making them more flexible in visual attention regulation. Elite male basketball players, as compared to amateur basketball players, are able to preserve their cognitive performance in tests probing visuospatial attention and decision-making even after an exhaustive bout of acute physical exercises (Mancı et al., 2023). Although the analysis of the number of players did not provide meaningful results, the relationship between the ball handler’s movement speed and motion decisions is significant.

Previous studies have shown that as the number of players on the field increases, players’ cognitive load significantly increases, affecting both decision-making time and accuracy (Vaeyens et al., 2010). In this study, the complex interaction between visual attention and motion decision-making was revealed by analyzing the performance of ball handlers under different speed conditions. Specifically, the average running speed and total running distance not only provide information on players’ physical condition but also offer insights into their decision-making ability at critical moments. At high speeds, players’ visual attention may be disrupted, leading to neglect of the surrounding environment, thus impacting decision-making quality. Research has shown a significant correlation between visual attention and sports decision-making. During high-speed sports, players tend to rely more on intuitive decision-making, which may lead to misjudgment. The results of this study confirmed this finding, showing that the accuracy of intuitive decision-making was significantly lower than that of cognitive decision-making (P < 0.001). Additionally, the ball handler did not make any shooting decisions while running at high speeds (7.98–12.62 m/s), providing important insights for practical training. This suggests that other offensive methods may be more effective than shooting in high-speed situations.

Spitz et al.’s (2018) research suggests that there is no significant difference in the accuracy of football referees when reviewing plays in slow motion compared to real-time decisions, and slowing down video speed does not improve decision accuracy, possibly due to a loss of authenticity in the real-time environment. In contrast to Spitz et al.’s (2018) findings, this study supports Lorains’ viewpoint. Lorains, Ball & MacMahon (2013) research explains why expert players make more accurate decisions at high speeds: faster speeds better align with the dynamic, time-sensitive decision-making environment in sports, meeting the cognitive processing needs of players. This is reflected in the experimental data: as speed increases, decision accuracy significantly decreases, particularly for shooting decisions. This decline may be attributed to the heightened cognitive load and attentional demands imposed by faster speeds. According to cognitive load theory (Ping, 2019), excessive processing demands can overwhelm working memory, impairing decision-making efficiency.

However, the expert group still showed significant advantages over the novice group under high-speed conditions, indicating that the fast-moving competitive environment places higher demands on players’ attention allocation and timely decision-making, further emphasizing the crucial role of visual attention in sports decision-making. Therefore, coaches and practitioners should consider incorporating high-speed visual tracking drills-such as dynamic target tracking, occlusion-based recognition tasks, and sport-specific decision scenarios-into training routines. These tasks should emphasize contextual realism, requiring players to respond under temporal constraints that closely mimic actual gameplay.

This study found the promoting effect of multiple object tracking training on sports decision-making ability, underscoring the close relationship between visual attention and sports decision-making. However, research on how multiple object tracking training affects the brain mechanisms of players is still insufficient. The expert group includes only players from one university-level league, the study exclusively included female participants, which restricts the generalizability of the findings to male players, the unbalanced number of decisions at different speeds, which reduces the diversity of expertise and limits the generalizability of the results. Future studies should integrate brain imaging technology to further explore the neural adaptive changes caused by training, providing a scientific basis for training strategies aimed at enhancing players’ decision-making abilities.

Conclusions

This study shows that expert basketball players display superior visual attention and decision-making in moderate-to-high complexity settings, extending dynamic sport cognition theory. The absence of differences at 5°/s and 15°/s suggests task boundary effects. Future work should examine player position and gender to refine perceptual–cognitive training and theoretical models.

Supplemental Information

Supplemental Information 1 Raw data.

Additional Information and Declarations

Competing Interests

The authors declare that they have no competing interests.

Author Contributions

Qifeng Gou conceived and designed the experiments, performed the experiments, analyzed the data, prepared figures and/or tables, authored or reviewed drafts of the article, and approved the final draft.

Sunnan Li conceived and designed the experiments, performed the experiments, authored or reviewed drafts of the article, and approved the final draft.

Human Ethics

The following information was supplied relating to ethical approvals (i.e., approving body and any reference numbers):

This work was approved by the regional ethics committee of the Northwest Normal University (No. NWNU-20230301).

Data Availability

The following information was supplied regarding data availability:

The raw measurements are available in the Supplemental Files.

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
