# Peer review of "The influence of basketball players’ tracking speed ability on sports decision performance"

_PeerJ, doi:10.7717/peerj.20266_

## Round 0.1 · original submission · Major Revisions

**Language Note:** The review process has identified that the English language must be improved. PeerJ can provide language editing services - please contact us at [email protected] for pricing (be sure to provide your manuscript number and title). Alternatively, you should make your own arrangements to improve the language quality and provide details in your response letter. – PeerJ Staff

Reviewer 1 ·

Basic reporting

Language and Style

The manuscript is generally written in understandable English, but several sections would benefit from stylistic refinement and more precise academic phrasing (e.g., lines 16–37). A professional English language edit is recommended to enhance clarity and scientific tone.

Introduction

The introduction requires a more academically structured narrative. Currently, it includes multiple contextual elements without clearly identifying a specific research gap. There is no unified theoretical grounding or justification of the study as a response to an identified gap in the existing literature. Given that the topic is not entirely novel, the authors should explicitly state the added value and innovative contribution of the current study. Moreover, line 43 lacks proper referencing, and line 50 would be more accurate if “stimulus” replaced “attention” .

Definition of Terms

Key terms such as “passing intuition” and “passing cognition” are used frequently, yet they lack clear, operational definitions. Given their central role in the study, the authors should provide precise definitions supported by relevant literature from sport psychology and cognitive science .

Experimental design

The study aligns well with the journal’s scope and applies the expert–novice paradigm appropriately. The combination of MOT testing and a video-based decision-making task is conceptually relevant. However, the methodology section lacks important details that are essential for transparency and replication:
• The authors do not report the psychometric characteristics (e.g., validity and reliability) of the applied tools, particularly for the MOT task and the sports decision-making task.
• Details about adaptation sessions, participants’ prior experience with MOT tasks, and standardization of experimental conditions are missing and should be clearly stated .
• The MOT task uses fixed object numbers and speeds (2–6 objects, 5–15°/s), but no adaptive (e.g., staircase) method was employed. This decision should be either justified or explicitly listed as a limitation.
Some visual materials (e.g., Figure 2) appear very similar to those in another submitted manuscript by the same authors (Study on the correlation between multiple object tracking ability and eye-tracking characteristics in sports decision making among basketball players), raising concerns about redundancy

Validity of the findings

Statistical Analysis

The data analysis includes appropriate use of ANOVA, but the following issues need to be addressed:
• There is no information regarding normality testing for the dependent variables. This is essential for justifying the use of parametric statistics. If assumptions are violated, non-parametric alternatives should be considered.
• The authors do not report effect sizes for post hoc tests, which are necessary to evaluate the practical significance of the findings.

Limitations

The limitations section should be substantially expanded. Suggested additions include:
• The expert group includes only athletes from one university-level league, which reduces the diversity of expertise and limits the generalizability of the results.
• The study’s ecological validity is constrained by laboratory conditions and the lack of motor responses, which do not fully reflect real-game decision-making.
• The use of fixed difficulty levels in the MOT task rather than an adaptive method also limits the sensitivity to individual performance differences.

Additional comments

This study addresses an interesting topic at the intersection of sport psychology, perceptual-cognitive expertise, and performance decision-making. However, the manuscript requires significant revision to meet PeerJ standards:
• The introduction should be restructured into a formal scientific rationale: theoretical background → research gap → purpose → novelty.
• The study’s contribution should be clearly articulated. At present, it reads as a confirmatory or replication study rather than a novel contribution.
• The conclusions (lines 448–456) are overly descriptive and largely reiterate the results. They should be reformulated to reflect interpretation and theoretical implications rather than repeating the findings .
• The literature review should be updated with more recent, peer-reviewed, and international sources, particularly from the last five years.

Reviewer 2 ·

Basic reporting

The manuscript is written in generally clear and professional English, and its structure follows the expected scientific format. Figures and tables are relevant, well-integrated, and support the main text effectively. After carefully reading the manuscript, I would like to commend the authors for their effort and contribution to the field. This is a well-designed and clearly structured study, with a logical progression from the introduction through to the conclusion. Overall, the manuscript meets the basic standards for scientific reporting and provides a meaningful basis for further discussion and application in basketball performance research.
However, some elements require revision or clarification in order to improve the overall clarity, coherence, and scientific impact of the paper. The introduction, for instance, would benefit from improved paragraphing and more concise formulation, particularly in the theoretical justification of the study. Furthermore, certain key terms (e.g., “intuition,” “cognition,” “perception,” “tracking speed”) are not always clearly distinguished, and I recommend defining them precisely early in the text to avoid confusion. The literature review, while citing relevant Chinese studies, would be significantly strengthened by the inclusion of more recent and diverse international references that reflect current developments in perceptual-cognitive training and sports decision-making.
These improvements would help ensure that readers can easily access scientifically grounded insights that are also practically applicable.

Experimental design

The experimental design is appropriate and well justified. The use of a dual-study format (MOT tracking and decision-making tasks) is a strength. The expert-novice paradigm is well applied, and the participant criteria are clearly described. Nevertheless, I recommend the following:
- Acknowledge the gender homogeneity (female-only sample) as a limitation to generalizability.
- Provide more detailed information on stimulus development and validation in Experiment 2. Although the authors describe a process for adjusting video difficulty, it would be useful to clarify the exact procedure used to ensure comparability across stimuli and speeds.
- It would also be helpful to briefly describe how potential confounding variables (e.g., prior video exposure, familiarity with game scenarios) were controlled or minimized.
These additions would strengthen the methodological transparency of the manuscript and help readers better evaluate the applicability of the findings.

Validity of the findings

The statistical methods are appropriate for the design. The authors correctly use repeated measures ANOVA with interaction effects and provide simple effects analyses and effect size (η²) values, which is commendable.
However, some statistical details are missing:
- The manuscript would benefit from more concise statistical reporting. I recommend reporting exact p-values and 95% confidence intervals where applicable.
- Given the complexity of the analyses in Experiment 2 (four-way ANOVA), visual aids such as interaction plots or graphs by condition would help clarify the findings.
The interpretations are generally accurate and in line with the results. However, some results are presented in too much numerical detail, which may overwhelm the reader.

Additional comments

This study contributes meaningfully to the field of perceptual-cognitive performance in team sports. The combination of multiple object tracking (MOT) and sports decision-making under varying speeds provides valuable insights. The use of first-person video is a methodological strength that enhances ecological validity.
To further improve the manuscript, I recommend the following:
- The abstract provides an informative overview of the study and clearly outlines the background, methods, and key findings. However, it contains an excessive amount of numerical detail, including means, standard deviations, and p-values for multiple conditions. This level of specificity is more appropriate for the Results section. I recommend reducing the statistical detail and focusing instead on summarizing the main trends and conclusions, to improve readability.
- Expand the discussion on how the findings could be translated into practical training recommendations (e.g., training visual tracking in high-speed decision contexts).
- Discuss cognitive load and attentional demands as potential mechanisms behind performance declines at higher speeds, especially for shooting decisions.
- Include a more explicit limitations section, addressing: the gender-specific sample; the unbalanced number of decisions at different speeds (e.g., no shooting trials at the highest speed); the relatively small number of high-speed trials in general.
Additionally, future work could explore eye-tracking, response time measurements, or visual attention training interventions.

·

Basic reporting

Dear Author,
First of all, congratulations on your valuable work.
In general, the study is well thought out and planned. The methodology section is also well designed. I believe that strengthening the introduction and discussion sections would further improve the quality of the study.
My more detailed comments and suggestions have been added as notes in the relevant sections of the PDF file.

Best regards,

Experimental design

The purpose and hypotheses of the study are adequately explained in the introduction. In addition,
the study' design was well planned and conducted. The experimental procedure was adequately described. My more detailed comments and suggestions have been added as notes in the relevant sections of the PDF file.

Validity of the findings

The results are related to the method used in the study and are sufficient. My more detailed comments and suggestions have been added as notes in the relevant sections of the PDF file.

Additional comments

I have nothing to add to the points I mentioned above. My more detailed comments and suggestions have been added as notes in the relevant sections of the PDF file.

---

## Round 0.2 · Major Revisions

The reviewers found the revised version of the manuscript improved but some substantial issues remains. I am confident that you could address these and improve overall clarity of the manuscript. The reviewers provided quite detailed suggestions that I would invite you to address carefully.

Reviewer 1 ·

Basic reporting

The authors have partially addressed the reviewers’ comments, but most of the key issues remain insufficiently resolved, which, in my opinion, continues to limit the overall scientific quality of the manuscript.

The manuscript has undergone a language edit and is now written in generally clear and understandable English. Several recent sources published within the last five years have been added, and the structure of the paper conforms to the journal’s formal requirements. However, the introduction is still weakly organized. It lacks a clear progression from theoretical background to research gap, objective, and the study’s contribution. Although the authors indicate that revisions were made, these improvements appear mostly superficial. The key terms “passing intuition” and “cognition” are still undefined. Simply stating that no precise definitions exist in the literature is not sufficient, because these concepts are central to the study and therefore require at least an operational clarification. In addition, the rationale for the hypothesis should be clearly grounded in previous research. The novelty of the approach and its relevance within the context of existing knowledge should also be explicitly articulated.

Experimental design

The study design is conceptually appropriate. The dual-part structure combining a multiple-object tracking task and a decision-making task is logically developed, and the expert-novice paradigm is applied correctly. Participant descriptions are adequate, and the clarification that none had previous experience with the MOT task is appreciated. Nevertheless, some essential methodological issues persist. The authors do not provide any psychometric characteristics, such as reliability or validity, for the instruments used, particularly for the MOT task. Citing previous studies, including their own, does not replace the need for direct reporting of these parameters. There is also no mention of any tests of normality, such as the Shapiro–Wilk test, although the authors claim that the data are normally distributed. Moreover, the fixed-speed MOT design does not incorporate any adaptive procedure, and this limitation is not acknowledged. In Experiment 2, the imbalance in action types across speed conditions (for example, the absence of shooting actions at 15°/s) suggests that the stimuli were not fully standardized, which may compromise the validity of comparisons between conditions.

Validity of the findings

In terms of statistical analysis, the authors correctly applied a four-way repeated measures ANOVA with follow-up simple effects analyses. They report η² values, which is commendable. However, effect sizes for post hoc comparisons are not reported, despite their importance for evaluating the practical relevance of the results.

The conclusions section is still too descriptive and should reflect broader theoretical implications rather than simply reiterating the results. The limitations section has been expanded, but remains rather general. It does not adequately address the potential influence of player position or gender differences in cognitive performance. Furthermore, the authors do not reflect critically on the lack of significant differences at 5°/s and 15°/s in the MOT task. This result is briefly explained as either too easy or too difficult, but it deserves a more nuanced interpretation, especially since it raises questions about the consistency and robustness of the findings.

Additional comments

While the authors have made partial improvements, the majority of the original concerns raised by the reviewer have not been adequately addressed. I encourage the authors to substantially revise the manuscript to enhance its clarity, methodological transparency, and theoretical contribution.

Reviewer 2 ·

Basic reporting

The revised version of the manuscript shows improved clarity, conciseness, and structure. The Introduction now offers sufficient theoretical background and clearly defines the study's position within the broader field of perceptual-cognitive research in sport. The addition of more recent and internationally relevant literature has strengthened the contextual framing. Minor stylistic and definitional improvements are still possible, but do not hinder comprehension. While the overall presentation is improved, some of the statistical results remain quite dense and could benefit from more concise reporting or visual support.

Experimental design

Methodology is sound and well justified overall.

Validity of the findings

The statistical analyses are appropriate and well justified. However, the Results section remains overly detailed, which may reduce readability. Including a few interaction plots or summary figures would make complex findings easier to interpret. Overall, the presentation could be improved, but this does not affect the overall quality of the study.

Additional comments

The Introduction, Discussion, and Conclusion sections have been strengthened, with improved explanations of practical applications and limitations.

While there is still room for improvement in the presentation of the results (e.g., use of visual summaries), these are minor editorial suggestions.

·

Basic reporting

Dear Authors,
I congratulate the authors on their revisions, which are mostly adequate.
I only have a couple of minor suggestions. The recommendations are available in the attached PDF file.
Best regards,

Experimental design

The revisions made are sufficient.

Validity of the findings

The revisions made are sufficient.

---

## Round 0.3 · accepted · Accept

Congratulations! You have successfully addressed the key issues highlighted by the Reviewers and the manuscript is now ready for publication.